# Nonsense mRNA suppression via nonstop decay

Joshua A Arribere[1]*, Andrew Z Fire[2]

[1]Department of Molecular, Cell and Developmental Biology, University of California, Santa Cruz, Santa Cruz, United States; [2]Departments of Pathology and Genetics, Stanford University School of Medicine, Stanford, United States

**Abstract** Nonsense-mediated mRNA decay is the process by which mRNAs bearing premature stop codons are recognized and cleared from the cell. While considerable information has accumulated regarding recognition of the premature stop codon, less is known about the ensuing mRNA suppression. During the characterization of a second, distinct translational surveillance pathway (nonstop mRNA decay), we trapped intermediates in nonsense mRNA degradation. We present data in support of a model wherein nonsense-mediated decay funnels into the nonstop decay pathway in *Caenorhabditis elegans*. Specifically, our results point to SKI-exosome decay and pelota-based ribosome removal as key steps facilitating suppression and clearance of prematurely-terminated translation complexes. These results suggest a model in which premature stop codons elicit nucleolytic cleavage, with the nonstop pathway disengaging ribosomes and degrading the resultant RNA fragments to suppress ongoing expression.

DOI: https://doi.org/10.7554/eLife.33292.001

## Introduction

Nonsense-mediated decay (NMD) (reviewed in [*He and Jacobson, 2015*]) is a translational surveillance pathway to mitigate deleterious products of premature stop codons. In NMD, recognition of an early stop codon destabilizes an mRNA (*Morse and Yanofsky, 1969*; *Baserga and Benz, 1988*; *Losson and Lacroute, 1979*). Foundational studies in *S. cerevisiae* and *C. elegans* revealed protein factors responsible for NMD (*Leeds et al., 1991*; *Hodgkin et al., 1989*; *Pulak and Anderson, 1993*). In the decades since, a large body of literature has highlighted similarities and differences in NMD between yeast and metazoans. For example, while both yeast and metazoan NMD involve a core set of three proteins (UPF1-3 in yeast, SMG-2–4 in metazoans), metazoans require additional proteins for NMD (e.g. SMG-1, –5, and −6). Additionally, *Saccharomyces cerevisiae* NMD is thought to occur predominantly through decapping and 5'>3' exonucleolytic degradation (*Muhlrad and Parker, 1994*), while studies across metazoans have implicated both exo- and endonucleolytic machineries (e.g. [*Lykke-Andersen, 2002*; *Lejeune et al., 2003*; *Gatfield and Izaurralde, 2004*; *Glavan et al., 2006*; *Huntzinger et al., 2008*; *Eberle et al., 2009*; *Lykke-Andersen et al., 2014*; *Schmidt et al., 2015*; *Ottens et al., 2017*]).

Although protective under many circumstances, the NMD pathway also contributes to pathological suppression of expression from numerous disease-causing mutations (about 11% of point mutations responsible for human disease [*Mort et al., 2008*]). Given the substantial pathological and protective significance of nonsense surveillance and the extensive degree to which the initial premature-stop-recognition machinery has been characterized, it is surprising that understanding of the downstream events leading to suppression of gene expression remains limited. Sources for the uncertainty include technical complications (e.g. the transient nature of RNA decay intermediates, loss of RNA decay machinery is lethal in many organisms) and differences in the NMD machinery between organisms (*He and Jacobson, 2015*). Further complicating the picture is the question of

*For correspondence: jarriber@ucsc.edu

**Competing interests:** The authors declare that no competing interests exist.

how degradative processes intersect with ongoing translation during NMD; as NMD is a translation-dependent process, at least one ribosome must be on the nonsense transcript when NMD initiates. Insight into these questions comes from a recent study in *Drosophila* where transient knockdown of nonstop decay factors stabilized nonsense mRNA fragments (*Hashimoto et al., 2017*).

Nonstop decay is a second translational surveillance pathway in which cells repress the activity of mRNAs lacking stop codons through both mRNA and protein decay mechanisms (*Frischmeyer et al., 2002*; *Bengtson and Joazeiro, 2010*). The dual mRNA and protein decay arms of this response are referred to (in aggregate) as nonstop decay. These confer a functional redundancy to nonstop decay that has hampered genetic approaches. Despite this redundancy, nearly two decades of work has illuminated some of the molecular players and mechanisms involved (reviewed in [*Klauer and van Hoof, 2012*]). Many of these players were initially implicated in nonstop from seminal work in *S. cerevisiae* (e.g. [*Frischmeyer et al., 2002*; *Bengtson and Joazeiro, 2010*; *Doma and Parker, 2006*]). Among these players is the SKI complex, thought to load a 3'>5' exonuclease on nonstop mRNAs, and a specialized ribosome rescue factor, *dom34/pelota*. Confounding metazoan analysis, mutations in *dom34/pelota* are lethal or sterile in two major metazoan systems (mammals and flies) (*Adham et al., 2003*; *Castrillon et al., 1993*). Contributions from a metazoan system where nonstop surveillance is active but nonessential would shed light on the evolutionary conservation, mechanisms, and roles of nonstop decay. Specifically, such a system could shed direct light on the intersection of RNA degradation and translation in metazoans.

Here, we report a metazoan system (*C. elegans*) where nonstop decay can be genetically manipulated. During our characterization of *C. elegans*' nonstop, we uncovered an unexpected link with NMD. Our subsequent results support a nonsense-mediated decay model in which recognition of premature stop codons results in cleavage of mRNAs at stop codons, generating truncated mRNAs which are further repressed by nonstop decay.

## Results

### *C. elegans* has nonstop mRNA decay

*C. elegans* is a genetically tractable animal system that has been instrumental in studying gene expression pathways (*The C. elegans Research Community, 2005*). Our reading suggested that *C. elegans* has a nonstop mRNA decay pathway, although previous efforts were unable to identify molecular players (*Parvaz and Anderson, 2007*). We set out to characterize nonstop decay in *C. elegans*.

First, we sought to determine the consequences for gene expression upon loss of all stop codons from a transcript. We selected the *unc-54* locus for our experiments as it has been extensively analyzed via molecular biology and genetics (*Brenner, 1974*; *Epstein et al., 1974*; *Dibb et al., 1985*; *Dibb et al., 1989*; *Moerman et al., 1982*; *Bejsovec and Anderson, 1988*; *Anderson and Brenner, 1984*). We started with a strain bearing a C-terminally integrated GFP lacking all but one stop codon (*Figure 1A*, [*Arribere et al., 2016*]). These animals were grossly wild type, with robust body wall muscle fluorescence and GFP localization at the periphery of muscle thick filaments, consistent with the known UNC-54 expression pattern. When we removed the last stop codon via CRISPR/Cas9, generating an *unc-54::gfp* locus lacking all stop codons *unc-54(cc2865)* (*Figure 1A*), the animals exhibited a profound Unc (uncoordinated) phenotype characteristic of strong loss-of-function alleles of *unc-54*. Examining RNA by RNA-seq for strains grown at 23C, we saw a ~6-fold loss of *unc-54* RNA for the nonstop allele (*Figure 1B*). To control for any secondary effect that muscular atrophy may have on *unc-54* levels in the nonstop strain, we also sequenced RNA from an *unc-54* mutant at 23C (*e1301*, which confers temperature-sensitive inactivation of UNC-54 protein at the nonpermissive temperature of 23C). Again, we observed a ~6-fold loss of the nonstop mRNA, suggesting the changes in *unc-54* mRNA level are not a secondary consequence of the Unc phenotype. At the protein level, we observed a > 100-fold loss of the UNC-54 nonstop protein (*Figure 1C,D*), and a lack of detectable GFP fluorescence, even under high-powered magnification. We thus conclude that a lack of stop codons destabilizes mRNA and is detrimental to protein expression in *C. elegans*.

There is reason to expect at least three independent mechanisms repressing expression of the *unc-54(cc2865)* nonstop allele:

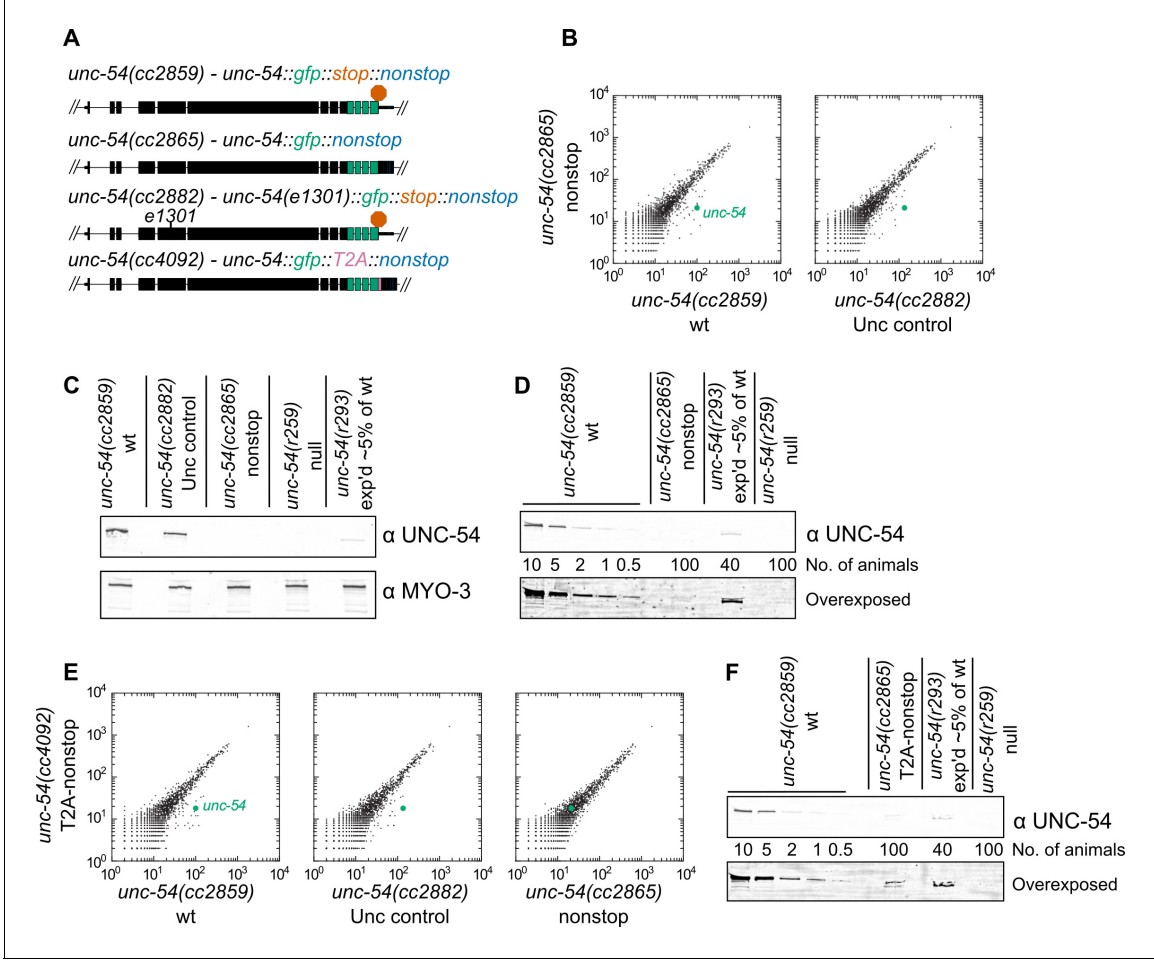

**Figure 1.** Loss of all stop codons is detrimental to mRNA and protein expression in *C. elegans*. (A) Diagram of alleles made via CRISPR/Cas9 and analyzed here. *cc2859* is an ancestor to the three other alleles. *e1301* is a temperature-sensitive Unc mutation, which was used throughout this figure as a control for Unc effects; T2A is a viral 'stop-and-go' peptide that releases upstream protein during translation elongation. (B) RNA-seq was performed to quantify mRNA levels in each indicated strain, with each dot representing read counts for a gene. Additional off-diagonal genes are collagens and vitellogenins, likely a secondary consequence of slow growth conferred by *cc2865*. (C) Immunoblot on an equal number of animals of each indicated genotype. *r259* is a ~17 kb deletion spanning much of *unc-54*; *r293* is a control allele exhibiting <5% of normal UNC-54 protein. Samples were split in half and probed for UNC-54 and MYO-3 (loading control) separately. (D) Immunoblot of UNC-54 with different titrations of animals loaded per lane. (E) RNA-seq as in (B), but with *cc4092* allele. (F) Immunoblot of UNC-54 as in (D), but with *cc4092* allele.

DOI: https://doi.org/10.7554/eLife.33292.002

The following figure supplement is available for figure 1:

**Figure supplement 1.** Results of candidate-based approach and genetic screen for nonstop suppressors.

DOI: https://doi.org/10.7554/eLife.33292.003

1. The first 28 3'UTR-encoded amino acids are sufficient to elicit ~20-fold repression at the protein level (***Arribere et al., 2016***). This is similar to the protein loss conferred by some 3'UTR-encoded peptides in yeast (e.g. ***Inada and Aiba, 2005***) and mammalian systems (e.g. ***Shibata et al., 2015***).

Additional repression would be expected to elicit nonstop decay via translation of the poly(A) tail at the level of:

1. nonstop protein decay, in which release of the nascent peptide is coupled to repressive mechanisms (e.g. [***Bengtson and Joazeiro, 2010***]), and
2. nonstop mRNA decay, in which ribosome stalling leads to degradation of the mRNA (***Frischmeyer et al., 2002***).

These multiple functionally redundant yet independent repressive mechanisms likely underlaid our failures to identify trans-acting factors through genetic screens or via candidate-based approaches (*Figure 1—figure supplement 1*). In light of our initial efforts, we sought to characterize nonstop mRNA decay independent of the protein degradation mechanisms.

We reasoned that release of the nascent peptide prior to translation of the 3'UTR and poly(A) tail would allow the UNC-54::GFP reporter protein to escape peptide repression acting from translation of the 3'UTR or poly(A) tail. Reporter protein levels, however, would still be tied to mRNA decay mechanisms. The 15 amino acid 'stop-and-go' T2A peptide co-translationally releases the upstream peptide, after which the ribosome resumes translation of downstream sequences. Indeed, a T2A peptide can liberate upstream protein from 3'UTR- or poly(A)-encoded protein repressive mechanisms (*Arribere et al., 2016*; *Sundaramoorthy et al., 2017*). We therefore integrated a T2A peptide downstream of *unc-54::gfp*, generating *unc-54(cc4092)* a 'T2A-nonstop reporter'. While this strain exhibited little difference in RNA levels compared to the T2A-less version (*Figure 1E*), protein expression was increased: animals had a faint but detectable GFP fluorescence, and UNC-54::GFP protein was detectable by immunoblot as 50–100-fold down (*Figure 1F*). These observations are consistent with the notion that the T2A peptide liberated UNC-54::GFP from some repression. We reasoned that a deficit in the nonstop mRNA decay pathway might be observed through phenotypic changes (an increase in movement and/or GFP fluorescence) in the T2A-nonstop reporter background.

## The SKI complex and *pelo-1* are required for nonstop mRNA decay in *C. elegans*

We used the *unc-54(cc4092)* T2A-nonstop reporter to screen for mutants that restored reporter protein expression. In ~55,000 mutagenized genomes, we isolated 17 recessive mutants with a similar phenotype: higher reporter expression and a mild but incomplete rescue of animal movement. Some isolates exhibited a weak rescue of egg-laying defects. It is notable that no mutant completely rescued the reporter, suggesting that the alleles recovered only partially restored UNC-54::GFP reporter expression.

Genetic mapping linked 14 mutants to a region of chrIV, and the remaining three mutants to a region of chrV (*Figure 2—figure supplement 1*). Genome-wide DNA sequencing identified mutations in the gene *skih-2(IV)* for the 14 mutants mapping to chrIV, and in the gene *ttc-37(V)* for the three mutations mapping to chrV (*Figure 2A,B*). Furthermore, a ~ 1 kb deletion in *skih-2* generated by CRISPR/Cas9 and crossed into the reporter strain restored UNC-54::GFP protein expression to a similar extent as the isolated mutants (*Figure 2A*). By sequence homology, *skih-2* and *ttc-37* resemble the yeast RNA helicase *ski2* and an associated factor *ski3*, respectively. This homology assignment for *ski2* is corroborated by missense mutations from our screen that hit amino acid residues conserved with other *ski2* sequence homologs (the sole missense mutation identified in *ttc-37* falls in a poorly conserved region of the protein) (*Figure 2A* and *Figure 2—figure supplement 2*). Altogether, our observations suggest *C. elegans* nonstop mRNA decay requires SKIH-2 and TTC-37, with these being the functional homologs of *ski2* and *ski3*.

Conspicuously absent from hits in our screen was a homolog of the *dom34/pelota* ribosome rescue factor. Loss of *dom34/pelota* modestly increases nonstop mRNA levels in multiple organisms (*Passos et al., 2009*; *Saito et al., 2013*). We identified *C. elegans r74.6* as a sequence homolog of *dom34/pelota*. Three lines of evidence suggest *r74.6* (hereafter *pelo-1*) is the functional ortholog of *dom34/pelota*:

1. *pelo-1* is the sequence ortholog of *dom34/pelota*, and has conserved functional residues known to be important for function in other systems (*Figure 2—figure supplement 3*).
2. While loss of either *pelo-1* or *skih-2* alone had only a mild effect on *C. elegans*' health at 23C, the *skih-2 pelo-1* double mutant was sterile at 23C (*Figure 2D*). This synthetic interaction is consistent with the idea that *skih-2* and *pelo-1* act in related processes. (The *skih-2 pelo-1* double mutant is weakly fertile at 16C, allowing for propagation of the strain.)
3. *pelo-1* deletion conferred a small, but reproducible increase in the T2A-nonstop reporter protein (*Figure 2E*). The small magnitude of this increase was likely too subtle to detect in our phenotypic screen, underlying our failure to isolate *pelo-1* alleles via forward genetics.

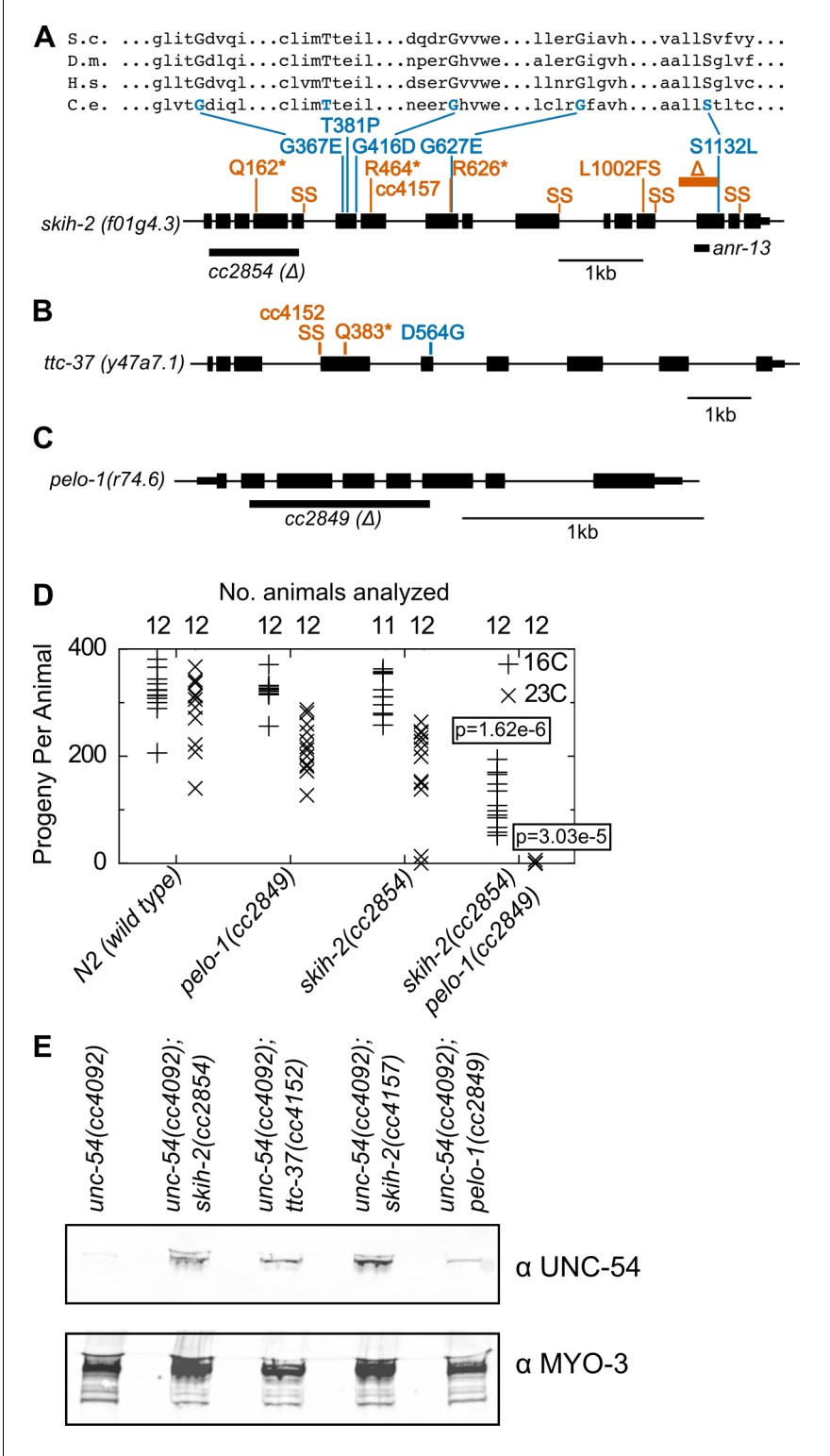

**Figure 2.** Protein factors required for nonstop mRNA decay. (**A**) Diagram of the *skih-2* gene, with mutations identified in the *cc4092* screen. Red indicates mutations expected to prematurely terminate SKIH-2 protein synthesis. SS is a mutation of a splice site. Blue indicate missense mutations, with multiple sequence alignment of these residues with *skih-2* homologs above. L1002FS is a frameshift mutation. 'S.c.' is *S. cerevisiae*; 'D.m.' is *D. melanogaster*; 'H.s.' is *H. sapiens* and 'C.e.' is *C. elegans*. A ~ 1 kb deletion introducing a premature stop codon made by CRISPR/Cas9 and used in subsequent characterization is indicated below the gene. See also *Figure 1—*

*Figure 2 continued*

*figure supplement 1*, *Figure 2—figure supplement 1*, *Figure 2—figure supplement 2*. (**B**) Diagram of the *ttc-37* gene, a *ski3* homolog, with mutations identified in the *cc4092* screen. Coloring as in (**A**). (**C**) Sequence homolog and candidate ortholog of *dom34/pelota*. ~700 bp deletion made by CRISPR/Cas9 and used in subsequent characterization is indicated below the gene. See also *Figure 1—figure supplement 1*, *Figure 2—figure supplement 3*. (**D**) Brood size analysis for the indicated strains. Each symbol represents brood size of one animal, with total number of animals analyzed indicated at top. See Materials and methods. p-Value from Mann Whitney U test compared with any other strain at the same temperature. (**E**) Immunoblot on an equal number of animals of the indicated genotypes.

DOI: https://doi.org/10.7554/eLife.33292.004

The following figure supplements are available for figure 2:

**Figure supplement 1.** Genetic mapping of suppressor strains.
DOI: https://doi.org/10.7554/eLife.33292.005

**Figure supplement 2.** Location of missense alleles on SKI-80S structure.
DOI: https://doi.org/10.7554/eLife.33292.006

**Figure supplement 3.** Multiple sequence alignment of *C.elegans* PELO-1 and sequence homologs.
DOI: https://doi.org/10.7554/eLife.33292.007

Together, our results are consistent with an important role for PELO-1 and the SKI complex in nonstop RNA decay.

## A nonstop decay mechanism conserved from *S. cerevisiae* to *C. elegans*

A model for PELO-1's and SKI's role in nonstop mRNA decay in *C. elegans* (which builds on results and models from homologous factors in *S. cerevisiae* (*Guydosh and Green, 2014*; *Guydosh and Green, 2017*; *Doma and Parker, 2006*; *Tsuboi et al., 2012*; *Frischmeyer et al., 2002*; *Passos et al., 2009*) is as follows (*Figure 3A*): A ribosome translates to, and then arrests at the 3'end of the mRNA. An endonuclease cleaves at the 5'edge of the ribosome, liberating the downstream stalled ribosome from the upstream mRNA. The downstream stalled ribosome is rescued with the help of PELO-1, and the SKI complex facilitates clearance of the 3'tail of the upstream mRNA fragment through its interactions with the 3'>5' exosome. As a trailing ribosome elongates to the 3'end of the mRNA fragment, the cycle repeats itself and the mRNA is degraded through recursive rounds of nonstop decay.

We set out to test this model, by monitoring expression at multiple levels of the T2A-nonstop *unc-54(cc4092)* allele, in wild type as well as *skih-2*, *pelo-1*, and *skih-2 pelo-1* mutant backgrounds. To measure the effects of *skih-2* and *pelo-1* mutations on mRNA levels, we performed RNA-seq (see Materials and methods). Loss of *skih-2* increased RNA levels from the *unc-54* allele (7.12 ± 0.18-fold, *Figure 3B*; mean ± SD from two biological replicates), consistent with the idea that loss of SKI compromises nonstop RNA surveillance at the RNA level. Loss of *pelo-1* also increased RNA levels, albeit to a weaker extent (3.81 ± 0.35-fold, *Figure 3B*). The reason for the *pelo-1*-dependent increase of nonstop mRNA levels may be direct or indirect; one model is that loss of *pelo-1* causes an increase of ribosomes on nonstop mRNAs, protecting the message from cellular RNases. We note that the mechanism of *pelo-1*'s effects on the nonstop allele mRNA levels is likely at least partly SKI-independent as the *skih-2 pelo-1* double mutant exhibited an increase in RNA levels greater than either single mutant (19.4 ± 2.2-fold, *Figure 3B*).

We also monitored translation of the *unc-54* nonstop mRNA via ribosome footprint profiling (Ribo-seq). A ribosome in the act of translating an mRNA will protect ~28–30 nt of mRNA upon RNase digestion in *C. elegans* (*Ingolia et al., 2009*). A ribosome stalled at the 3'edge of an mRNA would contain a partial or empty A-site, and thus protect a shorter ~15–18 nt mRNA fragment (*Guydosh and Green, 2014*). To capture both populations of ribosomes, we performed Ribo-seq of both size lengths (Materials and methods).

The level of translation (as assessed by 28-30nt footprints) in each strain mirrored the increases observed by RNA-seq (*Figure 3C*). However, there was a notable increase of 15-18nt Ribo-seq fragments upon loss of *pelo-1* (*Figure 3D*). The increase in 15-18nt Ribo-seq footprints in a *pelo-1* mutant was greater than would be expected by changes in either RNA-seq or the level of 28-30nt Ribo-seq footprints (*Figure 3E*), and was not observed with intermediate sized (19-22nt) Ribo-seq

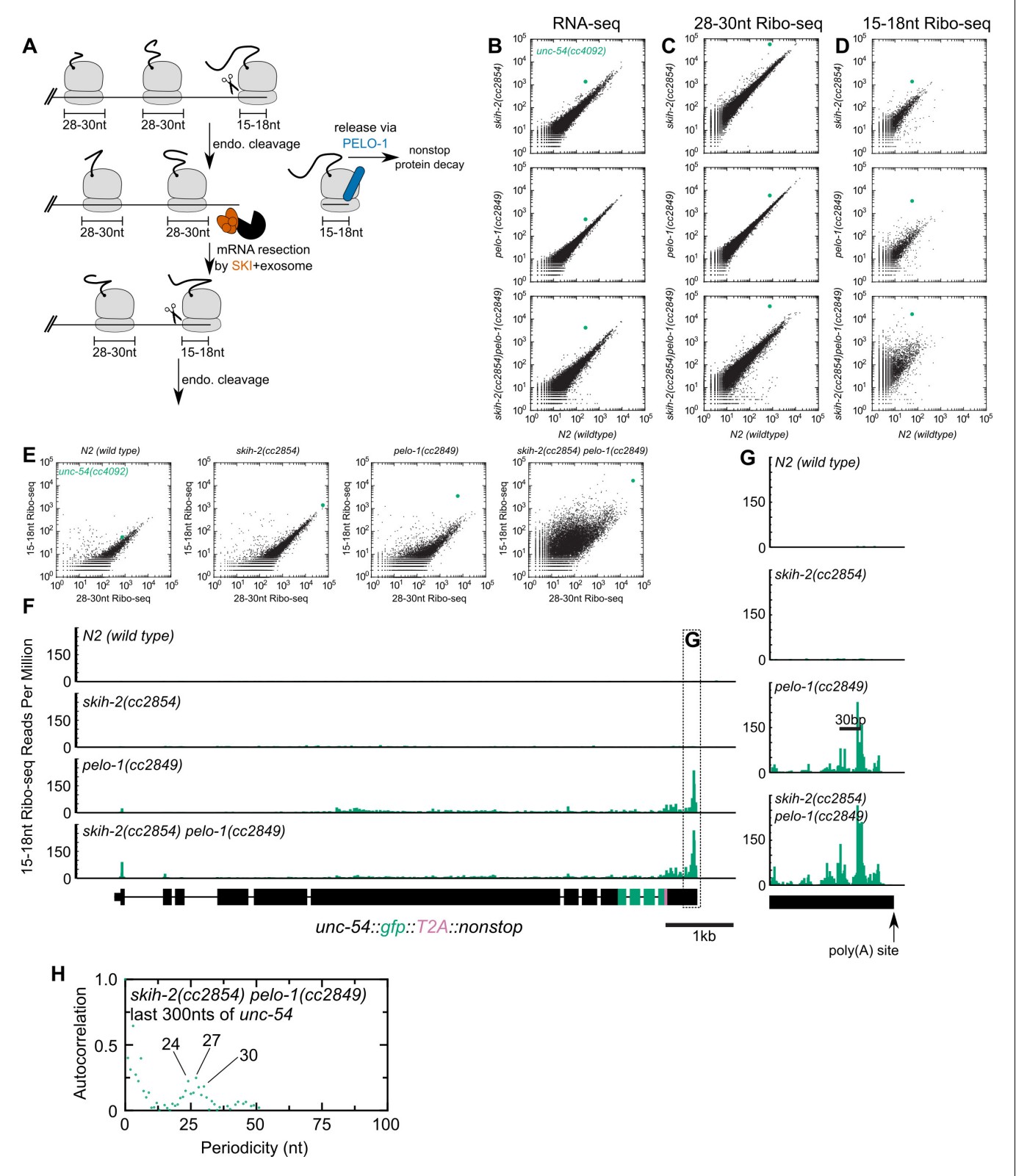

**Figure 3.** Analysis of gene expression during nonstop mRNA decay. (**A**) Model for nonstop decay, as described in the text. Briefly, ribosomes elongate to the end of an mRNA with no stop codons. This triggers endonucleolytic cleavage at the 5'edge of the ribosome by an unknown endonuclease. The resulting 3'end is a substrate for SKI (red) and the 3'>5' exosome, while the downstream ribosome is subjected to PELO-1-dependent rescue (blue) and nonstop protein decay. An upstream ribosome elongates to the 3'edge of the mRNA and the process repeats itself. Three ambiguities were difficult to

*Figure 3 continued on next page*

*Figure 3 continued*

represent in the model: (i) SKI can bind the 40S and the 3'>5' exosome and the order of binding events is not clear. (ii) It is unclear whether PELO-1 acts before or after the first endonucleolytic cleavage. (iii) It is unclear whether endonucleolytic cleavages occur successively, or simultaneously (e.g. cleavage may occur only once multiple ribosomes stall at the 3'end). (B) RNA-seq to quantify transcript levels. Each mutant strain is on the y-axis, and wild type is on the x-axis. Each dot represents read counts for a different gene, with *unc-54(cc4092)* locus highlighted (green). (C) Same as (B) for 28-30nt ribosome footprints. (D) Same as (B) for 15-18nt ribosome footprints. (E) Scatter plots of short (15-18nt) versus normal length (28-30nt) ribosome footprints for each strain background. This is an alternative way of displaying the data from (C) and (D). (F) Gene plot diagram of 15-18nt Ribo-seq reads mapping to *unc-54(cc4092)*, displayed according to where the 5'end of a read maps. Read counts are displayed per million uniquely-mapping reads. Dotted line shows area of focus in (G). (H) Pearson's autocorrelation of 15-18nt Ribo-seq reads mapping within 300nts upstream of the poly(A) site of *unc-54(cc4092)*. Similar results were observed with 100 or 200nts upstream of the poly(A) site. A biological replicate of these libraries yielded very similar results.

DOI: https://doi.org/10.7554/eLife.33292.008

The following figure supplement is available for figure 3:

**Figure supplement 1.** Justification for use of 15-18nt read lengths in Ribo-seq.

DOI: https://doi.org/10.7554/eLife.33292.009

footprints (*Figure 3—figure supplement 1*). The increase in 15-18nt footprints on the nonstop reporter is consistent with a model where the PELO-1 ribosome rescue factor is required for the efficient rescue of ribosomes with partial or empty A-sites. The increase in 15-18nt Ribo-seq fragments occurred across the entire *unc-54* nonstop transcript, but was especially prominent at the 3'end (*Figure 3F,G*). The sharpest accumulation of 15-18nt Ribo-seq reads was upstream of the poly(A) site, with a second peak another ~28–30 bases upstream. An autocorrelation analysis identified a periodicity of ~24–30 nt (*Figure 3H*). This accumulation of 15-18nt footprints phased by one ribosome width is consistent with a recursive model of nonstop decay (*Figure 3A*): endonucleolytic cleavage at the 5'edge of the leading ribosome generates another nonstop event as the trailing ribosome elongates to the resultant 3' terminus. The phased upstream accumulation of short footprints is consistent with recent reports in *S. cerevisiae* (*Simms et al., 2017*) and *S. pombe* (*Guydosh et al., 2017*), pointing to a conserved feature of nonstop decay.

Altogether our data support a model for nonstop decay conserved between *S. cerevisiae* and *C. elegans*: PELO-1 rescues ribosomes stalled on 3'-truncated RNA fragments, and SKI ensures efficient nonstop mRNA clearance.

## Hundreds of endogenous SKI/PELO substrates in *C. elegans*

The ability to genetically ablate nonstop machinery in *C. elegans* opens up the possibility of identifying endogenous SKI/PELO substrates. In the above analyses, we observed a population of endogenous mRNAs whose behavior mirrored that of the *unc-54(cc4092)* reporter: an accumulation of 15-18nt Ribo-seq footprints in *pelo-1* animals. We identified a population of 723 of these messages that exhibited reproducible accumulation of 15-18nt Ribo-seq footprints relative to 28-30nt Ribo-seq footprints specifically in a *skih-2(cc2854) pelo-1(2849)* mutant (*Figure 4A*, p<2.73e-6, DESeq (*Anders and Huber, 2010*), see also *Figure 4—source data 1*). We reasoned these mRNAs could represent mRNAs that produce RNA species that are targeted by SKI and PELO-1, hereafter referred to (for the sake of brevity) as 'endogenous SKI/PELO targets'.

In one particular case, we were able to identify a conserved nonstop target, *xbp-1*, the homolog of *S. cerevisiae hac1* (*Figure 4B*). *Xbp-1/hac1* is spliced by a tRNA endonuclease (*Shen et al., 2001*). In *S. cerevisiae*, this splicing is inefficient and at some rate, the cleaved *hac1* mRNA is translated, triggering nonstop decay (*Guydosh and Green, 2014*). We observed a peak of 15-18nt Ribo-seq reads at the precise nucleotides previously reported as 5' and 3' cut sites for the tRNA endonuclease within the *xbp-1* mRNA (*Figure 4C*). The 15-18nt Ribo-seq peak over the tRNA endonuclease site was only visible in strains with *pelo-1* deleted. These results are consistent with the idea that *C. elegans xbp-1* is inefficiently spliced and generates nonstop decay substrates, analogous to *S. cerevisiae hac1*.

It was possible that additional endogenous SKI/PELO targets would resemble the *unc-54(cc4092)* reporter or *xbp-1*: stop codon-less isoforms with truncated RNA fragments at or just upstream of the mRNA 3'end. This was not the case. Instead, endogenous SKI/PELO targets exhibited an accumulation of 15-18nt Ribo-seq reads at stop codons. For example, in addition to the tRNA

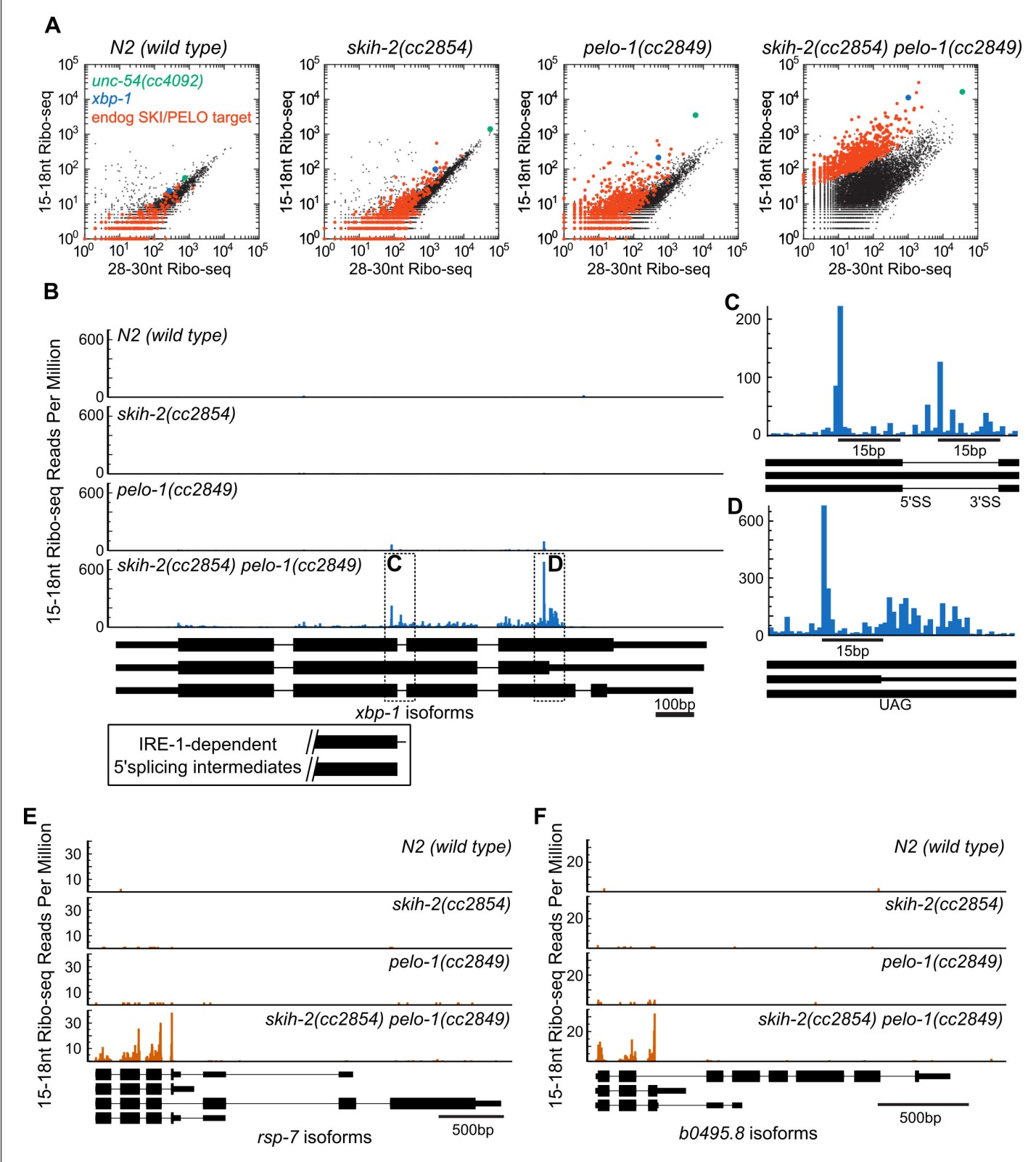

**Figure 4.** Hundreds of endogenous SKI/PELO targets. (**A**) Same data as in *Figure 3E*, with genes exhibiting consistently elevated 15-18nt Ribo-seq reads highlighted (red). One of these genes is *xbp-1* (blue), which is displayed in (**B**). (**B**) Gene plot diagram of 15-18nt Ribo-seq reads mapping to *xbp-1*, with reads displayed according to where their 5'ends map. Annotated *xbp-1* isoforms displayed below, and possible IRE-1-dependent splicing

*Figure 4 continued on next page*

*Figure 4 continued*

intermediates boxed. Read counts normalized to million uniquely mapping reads, as in *Figure 3F*. Dotted lines show regions of interest in (C) and (D). Note difference in scale in (C). (E) Gene plot diagram for *rsp-7*. (F) Gene plot diagram for *b0495.8*.

DOI: https://doi.org/10.7554/eLife.33292.010

The following source data and figure supplement are available for figure 4:

**Source data 1.** Gene-specific p-value for enrichment of *skih-2/pelo-1*-dependent 15-18nt Ribo-seq reads.

DOI: https://doi.org/10.7554/eLife.33292.012

**Figure supplement 1.** There are few endogenous SKI/PELO targets in *S. cerevisiae.*

DOI: https://doi.org/10.7554/eLife.33292.011

endonuclease site in *xbp-1*, we noted an accumulation of 15-18nt Ribo-seq reads overlapping the stop codon that would terminate the unspliced *xbp-1* reading frame (*Figure 4D*). Among endogenous SKI/PELO-targeted RNAs, several additional examples show representative cases where 15-18nt Ribo-seq reads accumulated over a stop codon (*Figure 4E,F*).

Both the number of endogenous SKI/PELO targets, and the location of 15-18nt Ribo-seq reads in *C. elegans* contrasted to previous results in *S. cerevisiae*. In *S. cerevisiae*, similar experiments with homologous factors (*ski2*, *dom34*) yielded comparatively few endogenous SKI/PELO substrates (*Guydosh and Green, 2014*). We thus reanalyzed existing datasets to identify nonstop targets in *S. cerevisiae* by their increased accumulation of 15-18nt Ribo-seq fragments relative to longer Ribo-seq fragments (*Figure 4—figure supplement 1*). Our analysis identified *hac1* (*xbp-1* homolog) and a handful of other mRNAs. Individual examination of these mRNAs failed to reveal accumulation of 15-18nt Ribo-seq reads over stop codons. We thus conclude that while at least one nonstop substrate is conserved (*hac1/xbp-1*), there is substantial difference in the number and nature of endogenous SKI/PELO-targeted RNAs between *S. cerevisiae* and *C. elegans*. We set out to better understand the link between stop codons and nonstop in *C. elegans*.

## Endogenous SKI/PELO substrates exhibit ribosomes stalled at truncated stop codons in *C. elegans*

To ascertain whether the relationship between stop codons and nonstop was more generalizable, we examined the *C. elegans* distribution of 15-18nt Ribo-seq reads genome-wide. In wild type and *skih-2* animals, we observed an approximately uniform distribution of 15-18nt Ribo-seq reads across the open-reading frame, and less in untranslated regions (*Figure 5A*). In *pelo-1* and *pelo-1/skih-2* mutant animals, we observed an increase in the abundance of 15-18nt Ribo-seq fragments just upstream of the stop codon (*Figure 5A*). The increase was greatest for reads with 5'ends mapping 14nt upstream of the first nucleotide of the stop codon, which corresponded to a stop codon in the A-site of the ribosome. Thus, stop codons accumulate 15-18nt Ribo-seq fragments in *pelo-1* mutants (~2.5-fold for *pelo-1(cc2849)* and ~5-fold for *pelo-1(cc2849) skih-2(cc2854)*). The endogenous SKI/PELO targets exhibited an even greater increase over these same positions (>13-fold for *pelo-1 (cc2849) skih-2(cc2854)*), consistent with our gene-by-gene analyses (*Figure 4B–F*). Thus, endogenous SKI/PELO targets accumulate several fold more stop codon-associated truncated fragments.

We initially expected ribosomes overlapping the stop codon to be substrates for eukaryotic Release Factor 1 (eRF1). In eukaryotes, eRF1 recognizes stop codons in the ribosomal A-site, facilitating translation termination and peptide release. We examined 15-18nt Ribo-seq fragments for clues as to their *pelo-1*-dependent accumulation. Examining individual stop codons we noted a trend which bore out genome-wide: in a *pelo-1* mutant, reads overlapping UAG and UAA codons tended to be truncated after the second or third bases, and reads overlapping UGA codons tended to be truncated after the second base (*Figure 5B*). It was extremely rare to find a full stop codon with an attached +1 nt (e.g. ...UAAN). These biases appear specific for stop codons: sense codons of the same nucleotide composition did not exhibit the same effects (*Figure 5B*, all codons in *Figure 5—figure supplement 1*). Truncation at a stop codon would be expected to yield an inefficient substrate for eukaryotic Release Factor-1-mediated translation termination: eRF1 protein makes contacts with all three stop codon nucleotides, as well as the +1 nucleotide (*Brown et al., 2015*). Thus, stop codon truncation would yield eRF1-resistant stalled ribosomes. While the precise substrate for PELO-1 is not known, previous work suggests that mammalian *pelota* can dissociate ribosomes with

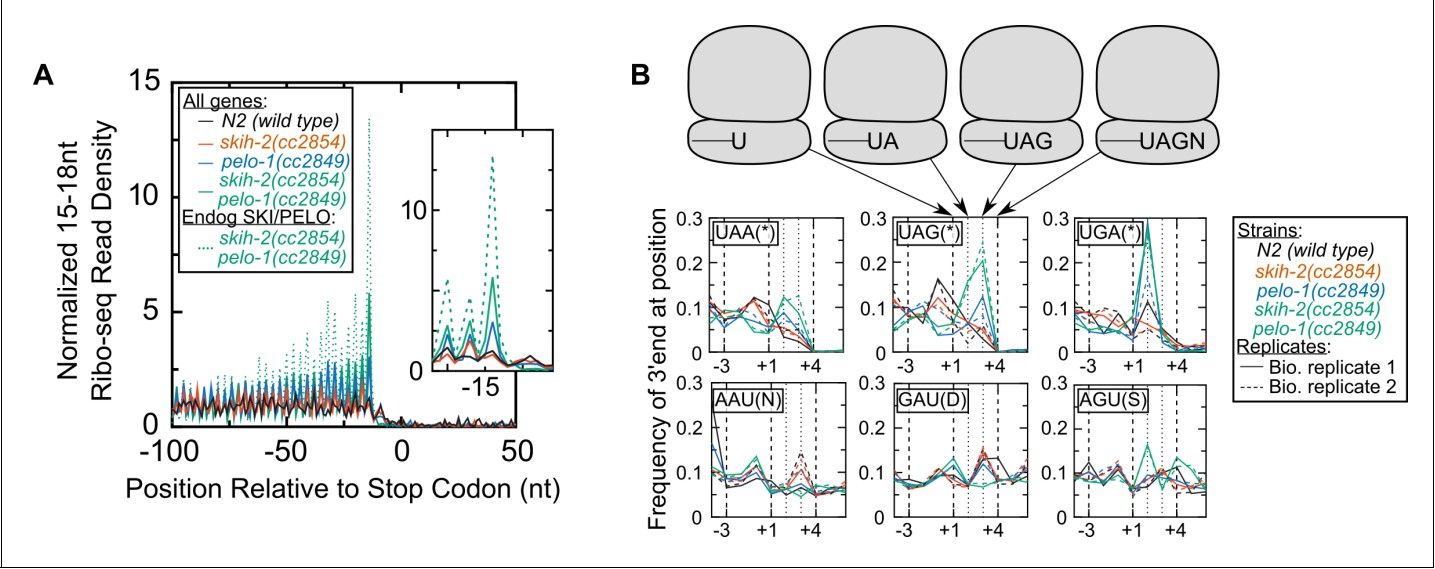

**Figure 5.** Ribosomes accumulate in the C-terminus and over truncated stop codons in *pelo-1*. (**A**) Metagene plot for all genes (solid lines), focusing on the C-terminus. Reads were assigned according to where their 5'ends map. Endogenous SKI/PELO targets (*Figure 4A*) are also shown (dotted line). Inset shows zoom of region just upstream of stop codons. (**B**) Metacodon plots showing where 15-18nt Ribo-seq reads terminate relative to nucleotides of each codon. For each nucleotide of each codon, we counted the number of times a 15-18nt Ribo-seq read terminated at that position (3' end), and normalized to positions upstream of the codon (see Materials and methods). Sense codons of similar nucleotide composition are shown as controls, and all codons in *Figure 5—figure supplement 1*.

DOI: https://doi.org/10.7554/eLife.33292.013

The following figure supplement is available for figure 5:

**Figure supplement 1.** Metacodon plot of 3'end locations for all codons.

DOI: https://doi.org/10.7554/eLife.33292.014

0, 1, 2, or three nts (including UAG) in the A-site (*Pisareva et al., 2011*). The increased detection of stop codon truncations in the *pelo-1* mutant is consistent with the idea that such complexes are substrates for PELO-1 rather than eRF1. Under this model, nonstop decay would be required to remove mRNAs truncated at their stop codon.

## Nonsense-mediated decay creates nonstop targets

Why do some mRNAs accumulate nonstop fragments to a much greater extent than the genome-wide average? We noted that among the endogenous SKI/PELO targets were several genes known to produce alternative mRNA isoforms known to be endogenous targets of nonsense-mediated mRNA decay, including: *swp-1*, *rsp-7*, *b0495.8*, *rsp-5*, *ubl-1*, *rsp-6*, *y57g11c.9*, *c12d8.1*, *asd-1*, *tos-1*, *hrpf-1*, *cyl-1* (*Barberan-Soler et al., 2009*). In each case, 15-18nt Ribo-seq reads accumulated preferentially at the stop codon of the mRNA isoform annotated as a nonsense decay target. Prior work suggested that nonstop and nonsense surveillance processes use distinct machineries with no mechanistic overlap (e.g. *Klauer and van Hoof, 2012*; *He and Jacobson, 2015*). However, the above observations led us to consider the possibility that nonstop decay substrates are generated as part of the nonsense-mediated decay pathway. We set out to test this possibility.

First, we tested whether activation of nonsense decay is sufficient to target an mRNA to nonstop. We inserted an early stop codon allele known to be a nonsense decay target to see if it could elicit nonstop decay. *unc-54(e1092)* is a premature stop codon (Qln >Stop) mutation known to elicit nonsense-mediated decay (*Dibb et al., 1985*). We introduced the *unc-54(e1092)* allele into the *skih-2 (cc2854) pelo-1(cc2849)* mutant. We observed substantial accumulation of 15-18nt Ribo-seq footprints at the *e1092* site (*Figure 6A*). As with nonstop decay, a second peak of 15-18nt Ribo-seq footprints appeared ~30 nt upstream of the *e1092* site (*Figure 6B*). The phased second peak is consistent with the idea that ribosomes stall at the *e1092* premature stop codon, triggering

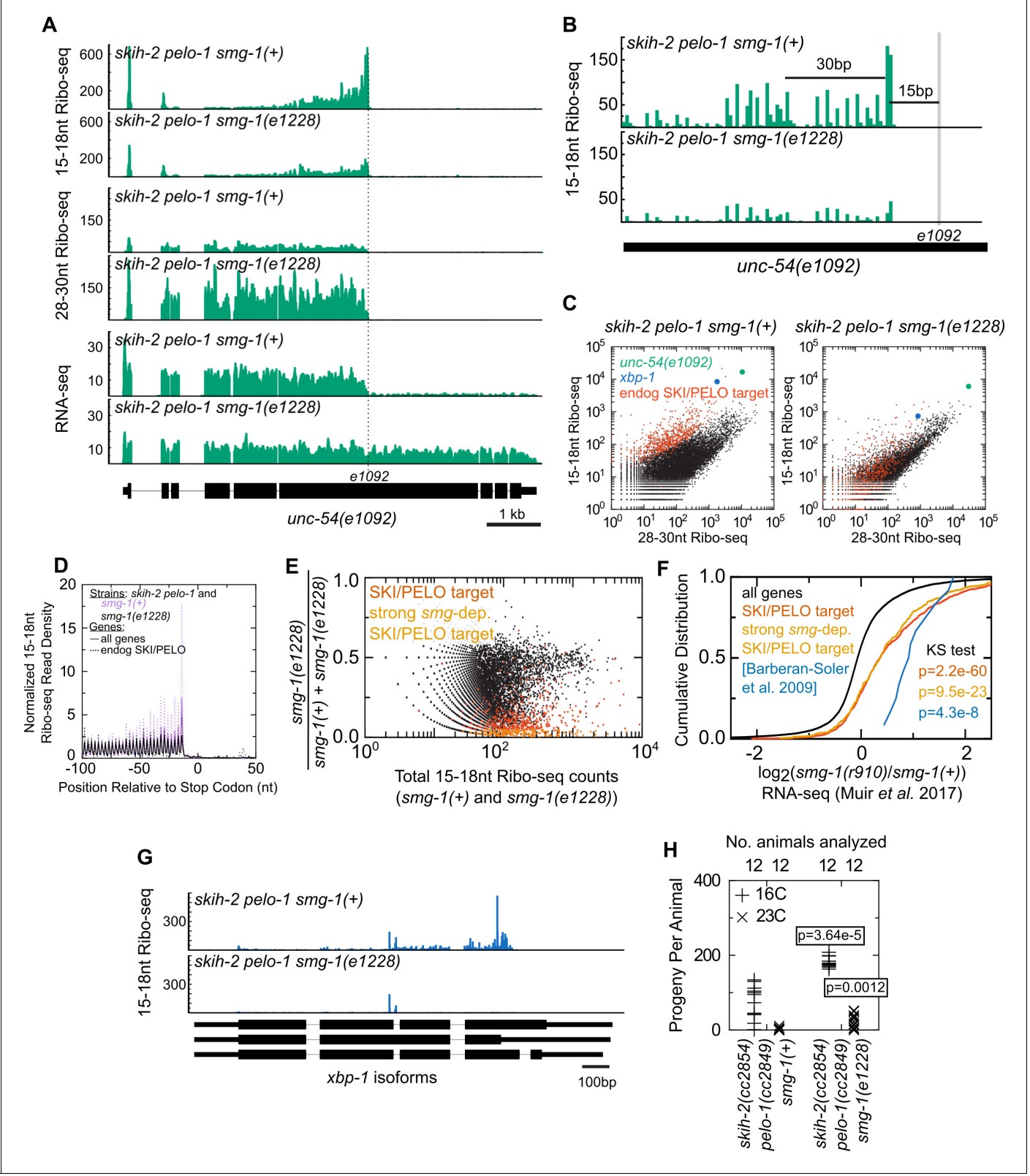

**Figure 6.** Nonsense decay generates nonstop substrates. (**A**) Gene plot diagram showing reads mapping to *unc-54(e1092)*, with position of *e1092* early stop codon indicated. Read counts are averaged over the entire sequence overlapping a read (not just the 5'end). All data in this figure were in the *skih-2(cc2854) pelo-1(cc2849)* double mutant background, abbreviated *skih-2 pelo-1*. (**B**) Zoom in of region upstream of *e1092*. Reads are displayed according to where their 5'ends map. (**C**) Genome-wide counts of Ribo-seq reads with and without *smg-1(e1228)*, with indicated genes highlighted. *Figure 6 continued on next page*

*Figure 6 continued*

'Endog. SKI/PELO' are the same genes identified in *Figure 4A*. (D) Genome-wide distribution of 15-18nt Ribo-seq reads with and without *smg-1* (*e1228*). Solid lines are all genes, and dotted lines are endogenous SKI/PELO targets, as identified in *Figure 4A*. Purple is the *smg-1(+)* strain, black is *smg-1(e1228)*. (E) X-axis shows the total number of 15-18nt Ribo-seq reads in a *skih-2/pelo-1* double mutant with and without *smg-1(e1228)*. Y-axis shows the fraction of those reads that come from the *smg-1(e1228)* mutant. Endogenous SKI/PELO targets are highlighted (red), and endogenous SKI/PELO targets with >95% of 15-18nt Ribo-seq reads derived from the *smg-1(+)* strain are highlighted in orange. (F) Log2 fold change in total mRNA levels as assayed by RNA-seq for genes shown in *Figure 1E*, using published datasets from *Muir et al., 2018*. Genes highlighted in blue are the 12 endogenous NMD targets identified in (*Barberan-Soler et al., 2009*) and described in the text. p-Value from Kolmogorov–Smirnov test relative to all genes (black line). (G) Distribution of 15-18nt Ribo-seq reads at *xbp-1* locus with and without *smg-1(e1228)*. (H) Brood size of *skih-2(cc2854) pelo-1 (cc2849)* with and without *smg-1(e1228)*. Brood size analysis performed as in *Figure 2D* and described in Materials and methods. p-Value from Mann Whitney U test. The *skih-2/pelo-1* double mutant was backcrossed to the *smg-1(e1228)* background as a control, and *smg-1(+)* homozygotes made by segregating away the *smg-1(e1228)* allele.

DOI: https://doi.org/10.7554/eLife.33292.015

The following figure supplement is available for figure 6:

**Figure supplement 1.** Bias against capturing A/T-rich RNAs in small RNA sequencing libraries.

DOI: https://doi.org/10.7554/eLife.33292.016

upstream endonucleolytic cleavage and a second nonstop substrate one ribosome upstream of the *e1092* site. This is similar to the pattern observed on the *unc-54(cc4092)* nonstop reporter. Thus, introduction of a premature stop codon is associated with an accumulation of reads in a manner consistent with nonstop RNA decay of the transcript.

We also performed RNA-seq on the same sample (*skih-2(cc2854) pelo-1(cc2849) unc-54(e1092)*) to examine *unc-54* RNA levels and species (*Figure 6A*). RNA-seq revealed a bimodal distribution of reads on *unc-54*: the *unc-54* mRNA upstream of *e1092* exhibited >10 fold more reads than the *unc-54* mRNA downstream of *e1092*. This observation is consistent with a model where SKIH-2 and PELO-1 are required for clearance of *unc-54(e1092)* mRNA. In the *skih-2/pelo-1* mutant, nonstop degradation of the truncated *unc-54(e1092)* open-reading frame is inefficient, and stalled ribosomes stabilize the degradation fragments generated by nonsense decay.

Second, we tested whether induction of nonstop at an early stop codon depended on known nonsense factors. *Smg-1* is important for nonsense-mediated decay, and loss of *smg-1* stabilizes nonsense mRNAs (*Hodgkin et al., 1989*). We thus generated and profiled the quadruple mutant *skih-2(cc2854) pelo-1(cc2849) unc-54(e1092) smg-1(e1228)*. In the *smg-1(e1228)* mutant there was a net reduction in 15-18nt Ribo-seq footprints on the *unc-54(e1092)* transcript (*Figure 6A*). This decrease is more significant after accounting for the higher levels of *unc-54(e1092)* in the *smg-1 (e1228)* mutant (*Figure 6C*). The distribution of RNA-seq reads on *unc-54(e1092)* also evened out in the *smg-1(e1228)* mutant. Thus loss of an intact nonsense-mediated mRNA decay pathway accompanies an apparent loss of targeting of the *unc-54(e1092)* nonsense mRNA via nonstop decay. We noted that a minority of 15-18nt Ribo-seq fragments overlapping *e1092* remained in the *smg-1 (e1228)* mutant. The source of this residual population of fragments is unclear, although possibilities include (1) residual activity of the nonsense-mediated decay pathway in the *smg-1(e1228)* background, whether from truncated SMG-1 protein or readthrough of *e1228*, (2) SMG-1-independent nonsense-mediated decay activity, or (3) a low level of nonsense-mediated decay-independent generation of nonstop fragments.

Third, we tested whether endogenous SKI/PELO targets were relieved from nonstop targeting upon loss of nonsense-mediated decay. We examined the behavior of endogenous SKI/PELO targets with and without *smg-1(e1228)*. As with *unc-54(e1092)*, the endogenous SKI/PELO targets exhibited a loss of 15-18nt Ribo-seq footprints relative to 28-30nt Ribo-seq footprints in the *smg-1(e1228)* mutant (*Figure 6C*). Thus, the nonsense decay pathway is generally required to elicit nonstop for several hundred endogenous SKI/PELO targets. Examining the distribution of 15-18nt Ribo-seq footprints revealed that *smg-1(e1228)* conferred a loss of 15-18nt Ribo-seq footprints specifically over the stop codon (*Figure 6D*). The loss of stop codon footprints was even greater for endogenous SKI/PELO targets. As with *unc-54(e1092)*, a residue of stop codon footprints persisted even in the *smg-1(e1228)* mutant (see preceding paragraph).

A gene-by-gene comparison of 15-18nt Ribo-seq read counts in *skih-2/pelo-1* with *smg-1(+)* or *smg-1(e1228)* revealed that the majority of endogenous SKI/PELO targets lose short footprints in

the *smg* mutant (*Figure 6E*). Consistent with the idea that SKI/PELO targets express transcripts that are degraded by NMD, steady-state mRNA levels of many of these genes increased in a *smg-1(r910)* mutant (using a previously published dataset from [*Muir et al., 2018*], *Figure 6F*). Many SKI/PELO targets exhibited a strong (>20-fold) *smg*-dependent change in 15-18nt Ribo-seq reads, and yet exhibited modest, if any, changes in total mRNA levels in the *smg-1(r910)* data. One explanation for this is that the short footprint assay is capable of identifying relatively minor transcript isoforms that are targeted by NMD relative to the background of 15-18nt Ribo-seq reads. If such isoforms are sufficiently rare, even a large fold de-repression in a *smg* mutant would be masked in RNA-seq by a lack of change in more abundant isoforms produced from the same gene. Consistent with this idea, we manually inspected several SKI/PELO targets and observed 15-18nt Ribo-seq reads accumulated at out-of-frame stop codons, although often without a transcript annotation that would explain such a translation termination event.

A few endogenous SKI/PELO targets still exhibited a high number of 15-18nt Ribo-seq fragments in *smg-1(e1228)*, among them *xbp-1*. Examination of *xbp-1* revealed that while the bulk of 15-18nt Ribo-seq reads over the early stop codon were lost in the *smg-1* mutant, some internal reads remained (*Figure 6G*). The reads that remained coincided with the known tRNA endonuclease cleavage sites in *xbp-1*. Thus, loss of the nonsense machinery specifically leads to loss of 15-18nt Ribo-seq reads over stop codons, but not for other classes of nonstop-targeting features.

Among the remaining ~20 largely *smg*-independent SKI/PELO targets, we observed untemplated adenosines in two genes: *y48e1b.8* (...TTACGGGTAAAA) and *f38e11.9* (...ACACTTCTCCCAAAAA), where ˆindicates the site of untemplated A's. Our ability to identify premature polyadenylation sites was likely hampered by an apparent bias in our short read libraries against A/T-rich sequences (*Figure 6—figure supplement 1*).

In the course of generating the above strains, we noted that introduction of the *smg-1(e1228)* allele into the *skih-2(cc2854) pelo-1(cc2849)* background improved animal health. This was most readily manifest in a partial rescue of fertility defects (*Figure 6H*). A model to explain this observation is as follows: A *skih-2/pelo-1* double mutant suffers as a result of an inability to deal with ribosomes stalled at the 3'end of cleaved RNAs. Many of these stuck ribosomes are generated via the nonsense-mediated decay pathway (*i.e.*, *Figure 6C*). Loss of *smg* activity in the *skih-2/pelo-1* background reduces the number of stalled ribosomes and thus improves animal health.

Thus, the activity of the nonsense-mediated decay pathway in *C. elegans* is coupled to the accumulation of nonstop RNA fragments at and upstream of the stop codon. The coupling of nonsense to nonstop (1) holds true for an engineered premature stop codon (*e1092*), (2) holds true for endogenous nonsense targets, and (3) depends on the known nonsense factor SMG-1.

## Discussion

### Lessons from *C. elegans'* nonstop decay

Several features have hampered the study of nonstop mRNA decay in vivo. By allowing a dissection of partially redundant aspects of nonstop decay and its consequences for target transcripts in the cell, we found that *C. elegans* provides a remarkable tool in understanding the process. Several conclusions stem from this analysis:

1. Nonstop mRNA decay is redundant. Because the cell responds to nonstop events through both protein and mRNA decay, two parallel pathways must be compromised to see an effect on a phenotypic reporter. Here we apply a tool (the T2A-based reporter) to study nonstop mRNA decay separate from nonstop protein decay. The T2A-based reporter enables both reverse and forward genetic studies of nonstop mRNA decay, and we expect that similar reporters may be useful in other systems where both mRNA and protein decay occur.

2. Removal of a ribosome from the 3'end of a nonstop mRNA is a critical cellular function. To-date there is no identified mutation or combination of mutations that yield complete derepression of a nonstop mRNA in *C. elegans* or *S. cerevisiae* (*Wilson et al., 2007*). The scarcity of alleles that completely de-repress nonstop mRNAs puts nonstop at a methodological disadvantage compared to nonsense-mediated decay where loss of any one of several factors (*upf1-3* in yeast, *smg-1–7* in *C. elegans*) yields approximately normal levels of protein expression.

3. There are hundreds of endogenous SKI/PELO targets in *C. elegans*. Here, the ability to knock-out nonstop decay factors provided an opportunity to define and study endogenous SKI/PELO targets. The nonessentiality of a seemingly core gene expression function, while surprising, has precedent: *C. elegans* tolerates some alterations to translation termination (*Wills et al., 1983*), nonsense-mediated decay (*Hodgkin et al., 1989*), RNA interference (*Tabara et al., 1999*; *Ketting et al., 1999*), and the splicing machinery (e.g. [*Run et al., 1996*; *Zahler et al., 2004*]). We expect that future studies in systems where nonstop is essential (i.e. flies, mammals) will benefit from the ability to knockout and study nonstop in *C. elegans*.

## A system to study SKI function

This experimental system has facilitated a characterization of roles for the SKI complex in mRNA decay. The *ski* class of genes was originally defined via the *superkiller* phenotype in yeast (*Toh-E et al., 1978*). Subsequently, it was found that some of the *ski* genes are required for nonstop decay (*Frischmeyer et al., 2002*). Outside of yeast, the structure and function of SKI may vary, with a directly parallel analysis complicated by the apparent absence of factors (i.e. *ski7*, whose function may be carried out by an alternative isoform of *hbs1* in humans, [e.g. *Kalisiak et al., 2016*]), low sequence conservation (i.e. *ski3*), and paralogy with other proteins (*i.e., ski2/mtr4*). Nonetheless, it is clear that complexes with similarity to the yeast SKI complex have important roles in a variety of RNA degradation processes in diverse systems (*e.g.,* [*Orban and Izaurralde, 2005*; *Branscheid et al., 2015*; *Hashimoto et al., 2017*]). Conversely, even within *S. cerevisiae* the role of the SKI complex in nonstop has not been universally accepted (*Inada and Aiba, 2005*). Here the isolation of *skih-2* (*ski2* homolog) and *ttc-37* (*ski3* homolog) from a genome-wide screen provides strong evidence that a comparable SKI complex exists in *C. elegans* and is required for nonstop mRNA decay.

The ability to genetically isolate and study SKI in vivo may help elucidate the specific mechanism by which SKI facilitates nonstop mRNA decay. A recent structural study found Ski2/3/8 p bound to the mRNA entry tunnel of the 40S small ribosomal subunit (*Schmidt et al., 2016*). What relationship, if any, Ski2/3/8 p at the entry tunnel has to nonstop mRNA decay is unknown. The combined application of genetics and synthetic biology to study nonstop in *C. elegans* offers a functional companion to these (and ongoing) structural studies of SKI and the ribosome. For example, here we isolated *skih-2(cc4142)*, a Gly >Glu missense mutation that mutates a conserved glycine packed at the interface of Ski2p and helix 16 of the small subunit rRNA (*Figure 2—figure supplement 2*). Analysis of *cc4142* and additional alleles from forward and reverse genetics may prove useful to interrogate the interface between SKI and the 40S to understand what role this interaction has, if any, in nonstop mRNA decay.

## Mechanistic implications for nonsense from nonsense/nonstop coupling

Previous work has suggested a functional separation of nonsense and nonstop (e.g. *Klauer and van Hoof, 2012*; *He and Jacobson, 2015*). The conceptual divide between nonsense and nonstop in the literature likely stems from the mechanistic dissimilarity of the pathways (one recognizes an early stop codon, the other an absence of stop codons), and the lack of overlap in molecular players identified by genetic screens (e.g. [*Wilson et al., 2007*; *Leeds et al., 1991*; *Hodgkin et al., 1989*; *Pulak and Anderson, 1993*]). However, recent transient knockdown work in *Drosophila* (*Hashimoto et al., 2017*) combined with our genetic and genomic analysis in *C. elegans* support a role for nonstop decay factors in clearing nonsense mRNAs. A simple model to reconcile earlier models with more recent observations is that nonstop suppresses a prematurely-terminated mRNA after it is committed to degradation by the nonsense-mediated decay machinery. This illuminates relatively unexplored steps in nonsense decay, namely the fate of the mRNA and mRNA:ribosome complexes after premature stop codon recognition.

An immediate implication of the model is that under normal circumstances nonstop decay would actively suppress the products resulting from a prematurely terminated open-reading frame. Thus the distribution of species in a nonstop-deficient background would allow for a more direct view of the products of the initial premature stop codon recognition events of nonsense. There are at least two models to explain the pattern of nonsense decay intermediates we observe:

1. Nonsense-mediated decay involves a ribosome-associated endonuclease that acts preferentially on or near premature stop codons. An analogy might be made with prokaryotic RelE, which binds and cleaves mRNAs in the ribosomal A-site. There is no sequence homolog of RelE in *C. elegans*. A candidate nuclease is *smg-6* which contains a C-terminal PIN domain distantly related to the nuclease PIN domain of RelE. Indeed, previous work suggests nonsense mRNAs are cleaved in the vicinity of premature stop codons (*Gatfield and Izaurralde, 2004*) with additional work implicating SMG-6 specifically (*Glavan et al., 2006*; *Huntzinger et al., 2008*; *Eberle et al., 2009*; *Lykke-Andersen et al., 2014*; *Schmidt et al., 2015*; *Ottens et al., 2017*).

2. Nonsense-mediated decay triggers 3'>5' exonucleolytic degradation, which then gives rise to nonstop substrates as 5'>3' translating ribosomes collide with an oncoming 3'>5' exonuclease. We disfavor this model for two reasons: (a) We would expect a more diffuse accumulation of 15-18nt Ribo-seq reads over premature stop codons. Instead, we observed a discrete and phased accumulation of 15-18nt Ribo-seq reads starting over the premature stop codon. (b) We would expect a population of intermediate size footprints (e.g. 19-28nt) that may arise from ribosomes translating to or colliding with 3'>5' exonucleases. We did not detect these intermediate fragments, although it is possible they are unstable and/or not detectable with our current Ribo-seq protocols.

Discerning between these and additional models will be a subject of future research.

We failed to detect coupling of nonsense to nonstop decay in *S. cerevisiae*. This may be because nonsense decay is fundamentally different in *S. cerevisiae* and *C. elegans*. For example, *S. cerevisiae* lacks evident homologs of four *smg* genes (among them *smg-5* and *smg-6*) required for nonsense decay in metazoans such as *C. elegans* and mammals. An immediate question for future research is whether other eukaryotes (*i.e.*, humans) exhibit nonsense/nonstop coupling as in *C. elegans*, or not as in *S. cerevisiae*. Answering this question may be technically challenging as *pelota* is essential in mammals (*Adham et al., 2003*).

## Broader implications for nonsense and nonstop

A large fraction of ribosomes stalled on RNA fragments in the *skih-2*/*pelo-1* mutant are derived from nonsense mRNAs. Together with phenotypic *smg-1* suppression of the *skih-2*/*pelo-1* sterility phenotype, our data suggest a major function of *C. elegans'* nonstop pathway is to clear nonsense decay intermediates. In light of this substantial connection, phenotypes currently attributed to insufficient nonsense or nonstop activity may be understood through their effects on the other pathway. Specifically, phenotypes from loss of nonstop factors may arise due to the persistence of truncated nonsense intermediates, and phenotypes from loss of nonsense factors may emerge from altered flux through the nonstop pathway. This model may serve generally useful for understanding how mutations in SKI homologs elicit trichohepatoenteric syndrome in humans (*Fabre et al., 2012*), loss of pelota/dom34 yields embryonic lethality in mice (*Adham et al., 2003*), loss of a nonstop protein decay factor (*lister*) yields neurodegenerative phenotypes in mice (*Chu et al., 2009*), and dysregulation of nonsense decay contributes to tumorigenicity in humans and mice models (*Wang et al., 2011*; *Popp and Maquat, 2015*).

Nonsense-mediated decay is often championed as a translational surveillance mechanism to mitigate production of truncated protein isoforms. However, nonsense-mediated decay requires translation to detect the premature stop codon, paradoxically generating a truncated protein to mitigate truncated protein production. The coupling of nonsense to nonstop may solve this problem: as nonstop is known to target both the RNA and nascent protein, the product from the initial round(s) of protein production may be degraded by nonstop protein decay. Under this model, the expression and toxicity of a prematurely truncated protein would be mitigated by nonstop decay. This has implications for genetic studies where nonsense alleles are used as loss-of-function alleles for a given protein of interest, as well as the ~11% of human inherited disease that occurs due to premature stop codon mutations (*Mort et al., 2008*).

# Materials and methods

## Strain construction and maintenance

All strains were derived from 'N2' (VC2010) background (*Brenner, 1974*) unless otherwise indicated. C. elegans were grown at 23 C on NGM plates seeded with OP50-1. *skih-2* and *pelo-1* mutants were grown at 16C. CRISPR/Cas9 was used to introduce edits (*Arribere et al., 2014*; *Paix et al., 2017*), and in each case multiple independent isolates were obtained with similar phenotypes to those shown. Mutant combinations were constructed via crossing as described in *Supplementary File 1*.

Brood size was measured by picking a single larvae to a freshly seeded small NGM plate. Every 24–48 hr, the animal was picked to a fresh plate until it stopped producing progeny. Its offspring were allowed to grow for a few days to make counting progeny easier. Plates exhibiting microbial contamination, or where the adult crawled off the edge, were excluded from the counts shown. For brood size calculations at 23C, we found that shifting *skih-2/pelo-1* double mutant larvae from 16C to 23C allowed that animal to produce a handful of progeny prior to becoming sterile. The resultant progeny spent their entire lives at 23C, and we analyzed their brood size as described above. No power analysis was performed to determine the number of animal broods to assay. The experimenter was blinded to genotype while picking animals and counting brood size.

## Immunoblotting

Western blotting was performed as described (*Arribere et al., 2016*). Briefly, all blotting and washing was done with Western Wash Buffer (1xPBS, 250 mM NaCl, 1.1% Tween 20). Blocking was done using 5% blotting-grade blocking reagent (Cat #1706404, Bio-Rad). The '5–6' antibody was used at 1:5000 to detect MYO-3 protein, '5–8' antibody was used at 1:5000 to detect UNC-54 protein (*Miller et al., 1986*), and a secondary Cy3 anti-mouse (Jackson Immunoresearch) was used at 1:500. Blots were scanned on a Typhoon Trio (Amersham Biosciences).

## RNA-seq

RNA-seq was performed essentially as described in ('RNA-seq2' *Arribere et al., 2016*). Briefly, animals were flash frozen in 50 mM NaCl and then ground with a mortar and pestle submerged in liquid nitrogen with ~3 x volume of frozen polysome lysis buffer (20 mM Tris pH 8.0, 140 mM KCl, 1.5 mM MgCl2, 1% Triton). RNA was extracted with trizol, and subjected to ribosomal subtraction with Ribo-Zero per the manufacturer's recommendations (Epicenter/Illumina). Ribosome-subtracted RNA was fragmented for 30' at 95C in 50 mM sodium carbonate buffer, pH 9.3. Fragmented RNA was gel purified and size selected (25-40nt) on a Urea-TBE acrylamide gel (15%). RNA fragments were eluted overnight in 300 mM NaAc pH 5.3, 1 mM EDTA and then precipitated. Library preparation continued with 3'PNK treatment, as described in *Arribere et al. (2016)*.

## Ribo-seq

Ribo-seq was performed essentially as described in *Arribere et al. (2016)*. Briefly, animals were flash frozen in 50 mM NaCl and then ground with a mortar and pestle submerged in liquid nitrogen. Animals were ground with ~3 x volume of frozen polysome lysis buffer. The resultant powder (~200 ul) was thawed by addition of 1 ml ice cold polysome lysis buffer with 100 ug/ml cycloheximide and kept on ice. Optical density (OD260) of lysates was measured, and 30U of RNase1 (Ambion) was added per OD unit. RNase1 digestion was allowed to proceed at room temperature for 30', and stopped by placing reactions on ice. Samples were loaded onto 10–60% sucrose gradients, and spun in an SW41 Ti rotor for 4.5 hr at 35,000 rpm (~150,000–200,000 rcf). RNA was isolated from monosome peaks by gradient fractionation, proteinase K digestion, and phenol/chloroform extraction. Size selection for full length (28-30nt) or truncated (15-18nt) footprints was done on a Urea-TBE 15% acrylamide gel using appropriate size standards (AF-MS-24 and AF-JA-267 (*Supplementary file 1*), respectively). After footprint isolation, RNA fragments were prepared for Illumina sequencing as previously described (*Arribere et al., 2016*).

## RNA- and Ribo-seq analyses

For *Figure 1*, libraries were sequenced on a MiSeq Genome Analyzer (Illumina, San Diego, CA). For *Figures 3* and *6*, libraries were sequenced with a NextSeq (Illumina).

We used ensembl release 83 (WBcel235) of the *C. elegans* genome. For strains bearing *unc-54* mutations (e.g. *unc-54(cc4092)*, *unc-54(e1092)*, etc.), we created custom versions of the genome with a modified *unc-54* locus and annotations. Prior to mapping, reads were collapsed to remove PCR duplicates, using the unique molecular adaptor (NNNNNN) ligated on the RNA with AF-JA-34. PCR duplicates consisted of no more than a few percent of reads for any given library. Collapsed reads were mapped to the genome with STAR (v2.4.2a) allowing for one mismatch. Uniquely mapping reads were size restricted, then assigned to genes according to *C. elegans*' ensembl release 83 annotations.

Pearson's autocorrelation (*Figure 3H*) was calculated as follows: For the 300nt upstream of the *unc-54* poly(A) site, the number of reads at each position was counted and stored as an array. We then shifted the array by x bases, and calculated pearson's correlation coefficient with the starting array. We performed this analysis for x = 0,1,...100, and plotted the correlation coefficient (*Figure 3H*).

To identify endogenous SKI/PELO targets, only Ribo-seq libraries from N2 and the double *skih-2/ pelo-1* mutant were considered (ignoring single mutant *skih-2* or *pelo-1* libraries). We tabulated gene counts for both 15-18nt and 28-30nt Ribo-seq libraries, restricted to reads with 5'ends mapping between [−12nt, −14nt] relative to the start and stop codon, respectively (*Ingolia et al., 2009*). We set a p-value cutoff of 2.73e-6 (0.05 divided by the number of annotated genes). Using DESeq, we identified genes enriched for 15-18nt Ribo-seq reads in the *skih-2/pelo-1* double mutant libraries relative to 15-18nt Ribo-seq reads in wild type and 28-30nt Ribo-seq reads in wild type or the double mutant.

The metagene analysis (*Figures 5A* and *6D*) was performed as previously described (*Ingolia et al., 2009*) such that each base of each transcript received the same weight, regardless of overall read count or transcript length. Unless otherwise indicated, reads were counted based on the position of the read 5'end. For the metacodon plots (*Figure 5B* and *Figure 5—figure supplement 1*), the frequency of read 3'ends terminating at each position was shown. To generate these plots, only reads with 5'ends mapping between [−12nt, −14nt] relative to the start and stop codon were considered (*Ingolia et al., 2009*). Loosening this restriction to include non-coding sequence-mapping reads produced similar results. For each nucleotide of each codon, we counted the number of times a 15-18nt Ribo-seq read terminated at that position. To normalize for codon usage and read coverage, we also counted the number of times a read terminated near that codon, up to nine bases upstream, and up to three bases downstream. We calculated the frequency of 3'ends at each position using the codon and the upstream three codons (ignoring the downstream codon so as to not skew stop codons because very few 15-18nt Ribo-seq reads terminate downstream of stop codons).

## Mutagenesis and suppressor screen

A large population of Unc animals were grown to ~L4, washed off plates, and incubated in 50 mM NaCl with mutagen at room temperature with rocking for 4 hr. For the PD2865 (unc-54::gfp::non-stop) screen, the mutagen was 50 mM EMS. For the PD4092 (unc-54::gfp::T2A::nonstop) screen, a cocktail of 25 mM EMS and 0.5 mM ENU was used. Animals were washed twice with M9, and allowed to recover for 24 hr on NGM plates with food. Mutagenized P0 animals were dissolved in sodium hypochlorite for ~7', leaving behind their eggs (mutant F1). Eggs were placed on an unseeded plate and larvae were allowed to starve and arrest as L1.~100 larvae were plated per small NGM, and 2 days later healthy F1 adults were counted to ascertain the number of genomes screened. At the F2/3 generation, animals were screened for increased movement and/or GFP. Only one isolate was kept per plate, ensuring independence of observed mutations.

## Suppressor mapping and variant identification

Suppressor loci were mapped similar to (*Doitsidou et al., 2010*), as depicted in *Figure 2—figure supplement 1*, and described here. Hawaiian *unc-54::mCherry* animals were made by CRISPR/Cas9, males isolated, then mated to each suppressor strain. Hawaiian (CB4856) is a wild *C. elegans* isolate with a SNP every 700–1300 base pairs. The F1 cross progeny were isolated and allowed to self for 2–3 generations, after which 20–50 GFP-positive animals were picked to a new plate. After a few

more generations, genomic DNA was isolated via proteinase K digestion and phenol/chloroform extraction.

30–60 ng of DNA was used to prepare deep sequencing libraries using the Nextera DNA Library Prep Kit (Illumina). Libraries were sequenced on a MiSeq Genome Analyzer (Illumina). Reads were mapped to the *C. elegans* genome (Ensembl Release 83) using bowtie2 (ver. 2.2.6 [*Langmead and Salzberg, 2012*]). Hawaiian-specific variants were identified using a high-coverage published dataset (*Thompson et al., 2013*) and GATK (*McKenna et al., 2010*), restricted to variants in uniquely-mapping regions. The list of high-confidence variants was used as a reference to assign reads in each sequenced backcrossed suppressor strain to either Hawaiian or N2. At the position of each variant, we then examined the fraction of reads derived from either Hawaiian or N2. We found that averaging variant frequency over a moving window of 100 variants reduced noise associated with sampling error of any given variant.

All 17 suppressor strains mapped in this manner displayed linkage to two loci. Visual inspection of reads (in IGV [*Robinson et al., 2011*]) revealed single nucleotide variants within one gene at each locus for 16 of the strains. A last variant (in PD4148) was found by a single read spanning a 476 bp deletion, which was confirmed by PCR and sequencing.

## Acknowledgements

We thank Sara Dubbury and Karen Artiles for advice on western blots, and members of the Fire lab for comments on the manuscript. We thank Rachel Green and Nicholas Guydosh for discussions on yeast *ski2* and *dom34*. This work was supposed by a NIH F32-NRSA fellowship (5F32GM112474-02) to JAA and R01 (NIH R01GM37706) to AZF.

## Additional information

### Funding

| Funder | Grant reference number | Author |
| --- | --- | --- |
| National Institutes of Health | R01GM37706 | Andrew Z Fire |
| National Institutes of Health | 5F32GM112474-02 | Joshua A Arribere |

The funders had no role in study design, data collection and interpretation, or the decision to submit the work for publication.

### Author contributions

Joshua A Arribere, Conceptualization, Resources, Data curation, Software, Formal analysis, Validation, Investigation, Visualization, Methodology, Writing—original draft, Project administration, Writing—review and editing; Andrew Z Fire, Conceptualization, Supervision, Funding acquisition, Investigation, Methodology, Writing—review and editing

### Author ORCIDs

Joshua A Arribere http://orcid.org/0000-0002-2467-7791
Andrew Z Fire http://orcid.org/0000-0001-6217-8312

### Decision letter and Author response

Decision letter https://doi.org/10.7554/eLife.33292.022
Author response https://doi.org/10.7554/eLife.33292.023

## Additional files

### Supplementary files

• Supplementary file 1. Oligos, Worm strains, and Plasmids used in this study Each table is a separate sheet; reagents available upon request.
DOI: https://doi.org/10.7554/eLife.33292.017

• Transparent reporting form
DOI: https://doi.org/10.7554/eLife.33292.018

## Major datasets

The following dataset was generated:

| Author(s) | Year | Dataset title | Dataset URL | Database, license, and accessibility information |
|---|---|---|---|---|
| Arribere J, Fire A | 2017 | Nonsense-mediated decay triggers nonstop mRNA decay in a metazoan | https://www.ncbi.nlm.nih.gov/sra/?term=SRP115527 | Publicly available at the NCBI Sequence Read Archive (accession no. SRP115527) |

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
