## [Decision Letter]

Thank you for submitting your article "Nonsense mRNA suppression via nonstop decay" for consideration by *eLife*. Your article has been favorably evaluated by James Manley (Senior Editor) and three reviewers, one of whom is a member of our Board of Reviewing Editors. The reviewers have opted to remain anonymous.

The reviewers have discussed the reviews with one another and the Reviewing Editor has drafted this decision to help you prepare a revised submission.

All three reviewers found the manuscript to tell a compelling and interesting story about a direct connection in *C. elegans* between NMD and NSD and all three are enthusiastic about publication in *eLife*. In addition to addressing the relatively minor concerns of the reviewers, detailed below, all three agreed that a somewhat more detailed analysis of the 723 targets identified in the SKI/PELO background should be performed. What fraction of these are true NMD targets (which correlate with RNA-Seq data from an appropriate NMD-background, for example) and what fraction represent prematurely polyadenylated genes, or others (such as Xbp1). Also, several reviewers suggested that experiments in a Smg6 delete background might add closure to the story, though they agreed that this is not necessary for publication. Once these changes are incorporated, the manuscript should be ready for acceptance and publication in *eLife*.

Reviewer #1:

Arribere and Fire offer strong evidence that the NMD pathway feeds into the NSD/NGD pathway in *C. elegans*. While this idea has been proposed before, the evidence presented here is very clear and appealing. While it would have been nice to see additional experiments to explore the role of other NMD factors with their novel short footprint assay, the story here is compelling.

While the data in Figure 5 and Figure 6 show the 723 endogenous targets of SKI/PELO are generally targets of the NMD pathway, it's possible that there could be other (minor) classes of targets (several sites remain enriched in short footprints in the *smg-1* animal). Did the authors check for premature polyadenylation on these genes, for example, by looking at the 3' end sequences of footprints that failed to align anywhere? It has been argued that sites of cryptic polyadenylation are also targets of SKI/PELO. Moreover, what fraction of the 723 genes did not have the stop codon as the major locus for short read enrichment?

In the third paragraph of the Introduction and the third paragraph of the subsection “The SKI complex and *pelo-1* are required for Nonstop mRNA decay in *C. elegans*”, the authors refer to *dom34/pelota* as a "release factor." This is incorrect since it is incapable of hydrolyzing the peptidyl-tRNA bond. It would be more accurate to call it a "ribosome rescue factor."

In Figure 3 and in the fourth paragraph of the subsection “A nonstop decay mechanism conserved from *S. cerevisiae* to *C. elegans*”, the authors describe ~30 nt periodicity. However, some peaks occur at shorter intervals. Similarly, the metagene plot for endogenous targets in Figure 5 shows a secondary peak only 18 nt behind the main stop peak. The authors should be cautious about specifying period length in the absence of more precise analysis. A short periodicity in the 15-18 footprint data might actually be expected for sequential ribosome-templated cleavages.

Figure 1 is generally unclear and very difficult to follow. The naming scheme for reporter constructs (*cc2859, cc2865*, etc.) is cryptic and would benefit from intuitive shorthand names. Some lanes on the gels are not described in the main text (*r259, r293*), making the figure a challenge to decipher. It's not clear how deletion of a 3'UTR in *r293* makes it a nonsense allele. Moreover, the use of *unc-54 (cc2882*) carrying the *e1301* temperature-sensitive Unc mutation isn't explained. Are any of the experiments carried out at the non-permissive temperature? Why is the right-hand panel of 1B or center panel of 1F required? Also, Figure 1 is never referenced in the text. It would appear that it should have been referenced just prior to 1E in the same sentence (panels should probably be reordered).

Why wasn't the third component of the SKI complex, SKI8, detected in the screen? Wilson et al. (2007) detected all three SKI complex components in their non-stop screen. Perhaps the authors could comment on this.

In the third paragraph of the subsection “The SKI complex and *pelo-1* are required for Nonstop mRNA decay in *C. elegans*”, the authors imply that loss of *dom34/pelota* should have given a large increase in mRNA levels and therefore have appeared in the screen. However, there are many cases in the literature where the effects are modest, consistent with results here, and the authors could note this.

Reviewer #2:

In their manuscript, Arribere and Fire describe the development of a genetic screen to identify genes involved in nonstop decay (NSD) in *C. elegans*. Similar to other organisms where NSD has been studied, they identify the SKI complex and *pelo-1* as being involved. During their transcriptome-wide search for NSD-substrates they discover that RNA substrates of the nonsense-mediated mRNA decay pathway (NMD) are also NSD substrates. More specifically, NSD degrades the upstream fragment produced after endonucleolytic cleavage at the premature termination codon that signifies NMD substrates.

The study is well-designed and well-described. I have only a few comments:

1) Based on knowledge from other organisms, the authors speculate that SMG6 is responsible for the endonucleolytic cleavage at the premature termination codon in NMD substrates in *C. elegans*. The authors should test if this is true to finalize their model.

2) The authors speculate that NSD may also be involved in the degradation of the upstream fragments produced by NMD in other organisms. Is this likely? There are some papers that have identified endonucleolytic cleavages by SMG6 transcriptome-wide at nucleotide resolution in human cell lines (Ottens et al., 2017; Schmidt et al., 2015; Lykke-Andersen et al., 2014). Often cleavage sites cluster downstream of the premature termination codon, which would mean that the upstream fragment would still contain a stop codon. This does not appear to be the case in *C. elegans* and should be considered by the authors.

3) The authors should also consider mentioning the study by Orban and Izaurralde, 2005, where it is implied that NSD is responsible for the degradation of upstream fragments produced after siRNA mediated endonucleolytic cleavage.

Reviewer #3:

In this manuscript from Arribere and Fire, the authors present novel data demonstrating a functional nonstop mRNA decay (NSD) pathway in *C. elegans*. In screening for mutants that suppressed NSD (using a very clever construct that deconvoluted the dual impacts of protein and RNA targeting systems), components of the SKI complex (specifically affect the RNA decay component) were discovered demonstrating conservation between *C. elegans* and other eukaryotes. The authors also identify an ortholog of ribosome rescue factor Pelota and demonstrate that genetic ablation of Pelota results in characteristic short ribosome protected fragments by ribosome profiling.

The key genome wide experiment is the observation of a substantial enrichment in short ribosome reads in the *skih-2/pelo-1* mutant animals for a large set of genes. When the distribution of short reads is evaluated, a large accumulation of reads is found directly at the stop codon with the 3' end of these reads located precisely at the +2 or +3 position of the stop codons (with the strong preference for the +2 position at UGA codons being particularly striking). Building on these observations, when the nonsense-mediated decay (NMD) pathway was disabled through mutation of *smg-1*, the signal of short reads at stop codons was lost. The authors reason that these short reads are generated by endonucleolytic cleavage during NMD (initiating perhaps through the initial cleavage at the stop codon following peptide release by eRF1/3) and the resulting upstream fragments are targeted and cleared by the canonical NSD pathway dependent on degradation by the SKI-exosome complex and ribosome rescue by pelota.

Both the data and conclusions presented here are sound and present a cohesive and interesting story. As detailed by the authors, in terms of the novelty, short ribosome footprints have previously been found enriched on truncated mRNAs when *ski2* and *dom34* (pelota) were deleted in yeast (Guydosh and Green 2014) and on NSD targets in the same background (Guydosh and Green, 2017). Endonucleolytic cleavage has been demonstrated during NMD in eukaryotes (Eberle 2008, Lykke-Andersen 2014) and this cleavage has even been shown to occur directly at stop codons (see Lykke-Anderson 2014 – Figure S2H). And, SMG-6 has been proposed to be the endonuclease responsible for cleaving NMD targets at stop codons and specificity for the +2 position of UGA codons has previously been identified (Schmidt et al., 2014).

Importantly, this paper does expand on the biological targets (in a new organism) for the exosome and ribosome rescue acting to clear RNA fragments generated by NMD, and this large cohort was not observed in yeast where NMD may not involve the actions of an endonuclease. And while this connection has been previously identified (Hashimoto, 2017), as referenced by the authors, the analysis here is more complete and in particular includes a genome-wide analysis and identification of targets.

In light of previous publications, it might be interesting to categorize the annotated endogenous SKI/PELO-1 targets into different groups that might include previously identified NMD targets (i.e. found in a screen for mRNA levels in a UPF1/SMG1 delete), prematurely polyadenylated mRNAs (if they have been documented in *C. elegans* or by looking for reads possessing iterated As at their 3' ends), or some other group.

---

## [Author Response]

Reviewer #1:Arribere and Fire offer strong evidence that the NMD pathway feeds into the NSD/NGD pathway in C. elegans. While this idea has been proposed before, the evidence presented here is very clear and appealing. While it would have been nice to see additional experiments to explore the role of other NMD factors with their novel short footprint assay, the story here is compelling.While the data in Figure 5 and Figure 6 show the 723 endogenous targets of SKI/PELO are generally targets of the NMD pathway, it's possible that there could be other (minor) classes of targets (several sites remain enriched in short footprints in the smg-1 animal). Did the authors check for premature polyadenylation on these genes, for example, by looking at the 3' end sequences of footprints that failed to align anywhere? It has been argued that sites of cryptic polyadenylation are also targets of SKI/PELO. Moreover, what fraction of the 723 genes did not have the stop codon as the major locus for short read enrichment?

We have updated our manuscript to include a more thorough analysis and discussion of the 723 SKI/PELO targets. As can be seen in the updated Figure 6, the vast majority of the 723 SKI/PELO targets exhibit *smg*-dependent accumulation of short ribosome footprints. We note, however, that many of these SKI/PELO targets exist as isoforms that are a small minority compared to other mRNAs expressed from their gene. Thus, for many SKI/PELO targets that exhibit a *smg-*dependent Ribo-seq accumulation, we see modest fold changes in total mRNA abundance by RNA-seq (Figure 6).

The remaining ~20 genes that have a largely *smg*-independent accumulation of short ribosome footprints would be good candidates for mRNAs that are endonucleolytically cleaved or that are prematurely polyadenylated. We found a few instances of the latter, though our ability to do so was hampered by the bias in our protocol against A-rich sequences (Figure 6—figure supplement 1). We have updated our manuscript with these additional analyses, and referenced them in the text (subsection “Nonsense-mediated decay creates nonstop targets”, sixth and eighth paragraphs).

In the third paragraph of the Introduction and the third paragraph of the subsection “The SKI complex and pelo-1 are required for Nonstop mRNA decay in C. elegans”, the authors refer to dom34/pelota as a "release factor." This is incorrect since it is incapable of hydrolyzing the peptidyl-tRNA bond. It would be more accurate to call it a "ribosome rescue factor."

Thank you for pointing this out--we have updated the manuscript to be more accurate.

In Figure 3 and in the fourth paragraph of the subsection “A nonstop decay mechanism conserved from S. cerevisiae to C. elegans”, the authors describe ~30 nt periodicity. However, some peaks occur at shorter intervals. Similarly, the metagene plot for endogenous targets in Figure 5 shows a secondary peak only 18 nt behind the main stop peak. The authors should be cautious about specifying period length in the absence of more precise analysis. A short periodicity in the 15-18 footprint data might actually be expected for sequential ribosome-templated cleavages.

We have performed an autocorrelation analysis, which yields a periodicity of ~24-30nt. We have updated our manuscript to include this (Figure 3, subsection “A nonstop decay mechanism conserved from *S. cerevisiae* to *C. elegans*”, fourth paragraph). Similar, very recent experiments have yielded periodic spacing of short ribosome footprints upstream of mRNA cleavage sites in diverse systems. It is notable that this periodic cleavage pattern is conserved, though with some difference in the reported periodicities (14 and 28nt in *S. pombe* (Guydosh et al., 2017), ~30nt in *S. cerevisiae* (Simms et al., 2017)).

Figure 1 is generally unclear and very difficult to follow. The naming scheme for reporter constructs (cc2859, cc2865, etc.) is cryptic and would benefit from intuitive shorthand names. Some lanes on the gels are not described in the main text (r259, r293), making the figure a challenge to decipher. It's not clear how deletion of a 3'UTR in r293 makes it a nonsense allele. Moreover, the use of unc-54 (cc2882) carrying the e1301 temperature-sensitive Unc mutation isn't explained. Are any of the experiments carried out at the non-permissive temperature? Why is the right-hand panel of 1B or center panel of 1F required? Also, Figure 1 is never referenced in the text. It would appear that it should have been referenced just prior to 1E in the same sentence (panels should probably be reordered).

We have updated Figure 1, as well as the main text (subsection “*C. elegans* has nonstop mRNA decay”, second paragraph) to make the figure more clear and easier to follow.

Why wasn't the third component of the SKI complex, SKI8, detected in the screen? Wilson et al. (2007) detected all three SKI complex components in their non-stop screen. Perhaps the authors could comment on this.

We don’t know why a *ski8* homolog was not identified in the screen. There are at least two non-mutually exclusive possibilities: (1) *ski8* is smaller than *ski2* or *ski3*. We expect our screen to identify longer genes more frequently than smaller ones because longer genes are easier to inactivate via mutagenesis. (2) *ski8* is known to have functions in meiosis (specifically DSB formation, see [Arora et al., 2004, Mol Cell]), and it is possible some essential phenotype of *C. elegans*’ *ski8* precluded its isolation in our screen. *C. elegans* has sequence homologs of *ski8*, and future work will test whether these have a phenotype with our nonstop reporter.

In the third paragraph of the subsection “The SKI complex and pelo-1 are required for Nonstop mRNA decay in C. elegans”, the authors imply that loss of dom34/pelota should have given a large increase in mRNA levels and therefore have appeared in the screen. However, there are many cases in the literature where the effects are modest, consistent with results here, and the authors could note this.

Consistent with the previous literature, we expect a modest increase in nonstop mRNA levels upon loss of *pelo-1*. We have clarified the manuscript to indicate this (subsection “The SKI complex and *pelo-1* are required for Nonstop mRNA decay in *C. elegans*”, third paragraph).

Reviewer #2: […] 1) Based on knowledge from other organisms, the authors speculate that SMG6 is responsible for the endonucleolytic cleavage at the premature termination codon in NMD substrates in C. elegans. The authors should test if this is true to finalize their model.

We have considered profiling a *smg-6/skih-2/pelo-1* triple mutant. The expectation from that experiment is the same whether *smg-6* is the causative nuclease or not (see *smg-1* experiment). We expect loss of any of *smg-1* through *smg-7* to yield the same effect on the distribution of short ribosome footprints. In any *smg* mutant the SMG machinery has lost the ability to detect (and presumably, cleave) premature stop codons. (An added complication is that *smg-6* null mutations are lethal in *C. elegans*, so only hypomorphs are available.) Definitive testing of SMG-6 as the nuclease will be done by us and others in future studies by other means (i.e., biochemistry and molecular biology).

2) The authors speculate that NSD may also be involved in the degradation of the upstream fragments produced by NMD in other organisms. Is this likely? There are some papers that have identified endonucleolytic cleavages by SMG6 transcriptome-wide at nucleotide resolution in human cell lines (Ottens et al., 2017; Schmidt et al., 2015; Lykke-Andersen et al., 2014). Often cleavage sites cluster downstream of the premature termination codon, which would mean that the upstream fragment would still contain a stop codon. This does not appear to be the case in C. elegans and should be considered by the authors.

Most studies on NMD cleavage sites have been conducted in cells with an intact nonstop decay system. Because we would expect SKI/PELO to clear cleavages within or upstream of the stop codon, one would expect to detect only those cleavages outside of ORFs (i.e., downstream of the stop codon). We expect that future studies in diverse systems (e.g., flies, mammalian cells) after SKI and PELO activity have been ablated will clarify whether this is the case.

Another group (working in flies), detected endonucleolytic cleavage near stop codons, and demonstrated that the upstream RNA fragment is subject to nonstop decay (Hashimoto et al., 2017). We note that at least some studies in mammalian cells have detected cleavage within the stop codon (e.g., Schmidt et al., 2015) similar to what we observe in *C. elegans*. Determining the full extent to which NMD is coupled to nonstop across organisms is beyond the scope of this current study, though we have referenced the above work in our manuscript (e.g., subsection “Mechanistic implications for nonsense from nonsense/nonstop coupling”, second paragraph).

Reviewer #3:[…] In light of previous publications, it might be interesting to categorize the annotated endogenous SKI/PELO-1 targets into different groups that might include previously identified NMD targets (i.e. found in a screen for mRNA levels in a UPF1/SMG1 delete), prematurely polyadenylated mRNAs (if they have been documented in C. elegans or by looking for reads possessing iterated As at their 3' ends), or some other group.

We have included a more in-depth analysis of the SKI/PELO targets in *C. elegans* (see first comment for review #1, updated Figure 6 and new Figure 6—figure supplement 1).